# MFDroid: A Stacking Ensemble Learning Framework for Android Malware Detection

**DOI:** 10.3390/s22072597

**Published:** 2022-03-28

**Authors:** Xusheng Wang, Linlin Zhang, Kai Zhao, Xuhui Ding, Mingming Yu

**Affiliations:** 1School of Cyber Science and Engineering, College of Information Science and Engineering, Xinjiang University, Urumqi 830046, China; wang_xs98@foxmail.com (X.W.); zhaokk@xju.edu.cn (K.Z.); xhding2021@163.com (X.D.); 2School of Software, Xinjiang University, Urumqi 830046, China; yumm0408@foxmail.com

**Keywords:** Android malware, ensemble learning, machine learning, static analysis, feature selection

## Abstract

As Android is a popular a mobile operating system, Android malware is on the rise, which poses a great threat to user privacy and security. Considering the poor detection effects of the single feature selection algorithm and the low detection efficiency of traditional machine learning methods, we propose an Android malware detection framework based on stacking ensemble learning—MFDroid—to identify Android malware. In this paper, we used seven feature selection algorithms to select permissions, API calls, and opcodes, and then merged the results of each feature selection algorithm to obtain a new feature set. Subsequently, we used this to train the base learner, and set the logical regression as a meta-classifier, to learn the implicit information from the output of base learners and obtain the classification results. After the evaluation, the F1-score of MFDroid reached 96.0%. Finally, we analyzed each type of feature to identify the differences between malicious and benign applications. At the end of this paper, we present some general conclusions. In recent years, malicious applications and benign applications have been similar in terms of permission requests. In other words, the model of training, only with permission, can no longer effectively or efficiently distinguish malicious applications from benign applications.

## 1. Introduction

Android is the mobile operating system with the highest market share in the world. As of December 2021, the market share of Android was as high as 70% [1]. As the number of Android users has risen in recent years, malware, such as financial losses and privacy disclosure, have become more common. In many Asian countries, the risk of being infected with malware is much higher. There are many app stores provided by various third-party vendors and many smartphones have been rooted. There are about 1.61 billion active mobile devices in China, of which, about 78.6% run Android as the operating system [2]. Therefore, from the perspective of Android information security, it is necessary to research malware detection technology and improve detection performance.

At present, there are two mainstream malware detection methods, static detection and dynamic detection [3,4]. The techniques involved in static detection include decompilation, reverse analysis, and static system call analysis. Static analysis does not need to run the application, it uses decompilation tools to perform lexical analysis, semantic analysis, etc., on the static code to extract features. APK files contain many features, such as permissions, API calls, signatures, network addresses, and hardware structures. All features can be used as a basis for judging whether an Android application is malicious. Another detection technique is dynamic detection, which places the application in a sandbox isolated from the outside world and observes the behavior of the application. Dynamic analysis involves analyzing the behavior characteristics of the application without disturbing the external software and hardware environment. If the behavior of the application is found to be inconsistent with the routine, it is judged as malware. Dynamic features are mainly extracted from the “behavior” that occurs when an Android application is running, and the most important of these features are the underlying system call information and network traffic interaction information. Every application requires the operating system to provide the most basic resources and services to achieve its functions. At the same time, due to the characteristics of Android system architecture, applications belonging to the application layer cannot directly interact with the operating system, so the application also needs to use system calls to perform certain tasks, such as file read, write, and open [5].

With the development of machine learning technology, more researchers have applied it to Android malware detection [6]. They extract features of Android application through static analysis or dynamic analysis and utilize machine learning classification models to detect Android malware. Although malware detection based on a single model has been clearly studied, the performance of each classifier varies due to differences in training data and feature selection algorithms. To overcome the shortcoming of single-model-based methods, an ensemble learning method is proposed. It combines multiple individual classifiers and usually achieves better generalization performance than a single model. Therefore, we built a stacking ensemble framework, called MFDroid, for malware detection. We collected 1664 real applications from 4 app markets in China to evaluate the detection performance of MFDroid. At the same time, we performed statistical analysis on our dataset. The results show that MFDroid proposed in this study is effective, which can significantly distinguish malicious applications from benign applications. The main contributions of this paper are as follows:We constructed a real-world Android malware dataset and conducted a comprehensive analysis for the dataset, showing the differences between malicious and benign applications.We combined the results of seven feature selection algorithms and fed them into the model for training. The results show that the F1-score of our method is significantly higher than other feature selection algorithms.We propose an Android malware detection framework based on stacking ensemble learning—MFDroid.

The remainder of this paper is organized as follows: Section 2 presents related work and the current state of research on dataset construction, static analysis methods, machine learning, and ensemble learning. The details of the dataset are described in Section 3. In Section 4, we present the proposed framework and method in detail, and then in Section 5, we analyze the experimental results on our dataset with our proposed method. Our conclusions are summarized in Section 6.

## 2. Related Work

Due to the increase of malware on smart devices, Android malware detection has become an important research topic, and extensive studies have been produced. Concerning the general process of Android malware detection, including dataset construction, static feature extraction, feature selection, machine learning methods, and ensemble learning methods, we review the related work in the field of malware detection.

### 2.1. Construction of Dataset

In recent years, the permissions requested by Android malware have been very close to benign applications. Older datasets can no longer meet the needs of accurate detection, so it is necessary to build a newer dataset for malware detection.

Arp et al. [7] presented a dataset of real Android applications and real malware, and they collected an initial dataset of 131,611 apps. The samples were collected between August 2010 and October 2012.

Allix et al. [8] constructed AndroZoo, an Android application dataset of over 17,854,943 apps, including Google play, each of which was analyzed by dozens of different antivirus products to mark malware.

Martín et al. [9] constructed a large comprehensive feature dataset consisting of 22,000 real malware and benign applications. Their work could help anti-malware tool researchers develop new mechanisms and tools for Android malware detection.

Given the early creation of the above datasets, it is not possible to conduct research on the latest Android application security situation. Therefore, we create a new dataset containing APKs collected during 2019–2020.

### 2.2. Static Analysis

Static analysis involves utilizing decompilation tools to extract features through lexical analysis, semantic analysis, etc., of applications. APK files contain many features, such as permissions, API calls, signatures, network addresses, and hardware structures, etc., which can be used as a basis for judging whether an Android application is malicious.

Li et al. [10] proposed a malware detection system SIGPID based on permission usage analysis, and developed three levels of pruning by mining permission data to identify important permissions that could effectively distinguish benign applications from malicious applications. SIGPID utilized machine learning-based classification methods to classify different families of malware and benign applications.

Tao et al. [11] proposed MalPat, an automatic malware detection system that automatically extracts key features from tens of thousands of applications to help the Android app market fight malware. Using statistical analysis techniques, they delved into different usages of permissions and APIs and revealed their ability to distinguish malware from benign applications. MalPat took full advantage of the characteristics of machine learning methods and used it earlier in the system architecture to learn the latent features of existing data.

Alazab et al. [12] proposed an efficient classification model that combined permission requests and API calls. Three different grouping strategies were proposed to select the most valuable API calls to maximize the likelihood of identifying Android malware applications. An in-depth analysis of different public and private packages, classes, and methods was performed to evaluate the effectiveness and accuracy of the proposed method when dealing with large datasets.

Although the above research uses the fine-grained feature of API call, it does not use Opcode, which is also a fine-grained feature, as one of the features. Some studies have shown that incorporating Opcode into the feature sets for experiments is helpful to understand the complexity and behavior of applications [13]. Therefore, this paper adopts three features: permission, API call, and Opcode.

### 2.3. Machine Learning

A machine learning classification model can further discover features and hidden rules between features and make predictions by fitting known data.

Cai et al. [14] proposed a novel feature-weighted-based Android malware detection scheme that combined the optimization of weight mapping functions and classifier parameters, called JOWMDroid. First, eight categories of features were extracted from the Android application package, and then the information gain was used to select a certain number of the most important features for malware detection. Next, initial weights were calculated for each selected feature through three machine learning models, and then five weight mapping functions were designed to map the initial weights to the final weights. Finally, the parameters of the weight mapping function and the classifier were jointly optimized by the differential evolution algorithm.

Mahindru et al. [15] proposed an effective and efficient web-based Android malware detection solution, MLDroid, capable of detecting malware applications based on permissions and API calls. They implemented different supervised, unsupervised, semi-supervised, and hybrid machine learning algorithms to train MLDroid, enabling it to achieve higher detection rates.

Jannat et al. [16] proposed a system to analyze and detect Android malware using machine learning. They solved problems in two ways—dynamic analysis and static analysis. The best results were obtained in dynamic analysis using the random forest (RF) algorithm, which is an extended variant of the decision tree (DT) algorithm. In addition, researchers used various datasets for static and dynamic studies.

Utilizing only one machine learning algorithm makes the experiment subject to skewed data, and the computational overhead is relatively large. In addition, it may be affected by the initial setting, is sensitive to noise, and cannot handle high-dimensional features well. Therefore, we used an ensemble learning approach to overcome the limitations of a single machine learning algorithm, thereby improving the accuracy of malware detection.

### 2.4. Ensemble Learning

Ensemble learning involves combining multiple weakly supervised models to get a better (and more comprehensive) strongly supervised model. The key idea of ensemble learning is that, even if a weak classifier gets a wrong prediction, other weak classifiers can correct the error.

Aboaoja et al. [17] proposed an ensemble behavior-based early evasion malware detection framework. The developed framework consists of three main stages, evasion behavior collection, correlation-based feature extraction and selection, and model building stage. The framework was able to efficiently identify complex malware behaviors using ensemble learning methods and make final decisions based on the results of the majority voting strategy.

Zhu et al. [18] proposed a stacking integration framework, SEDMDroid, to identify Android malware. Principal component analysis was performed on each feature subset to detect accuracy by retaining all principal components and using the entire dataset to train each base learner multilayer perception (MLP). Then, a support vector machine (SVM) was used as a fusion classifier to learn implicit supplementary information from the outputs of ensemble members and produce final predictions.

Idress et al. [19] proposed a new permission- and intent-based framework, PIndroid, to identify Android malware applications. PIndroid was the first solution to accurately detect malware using a combination of permissions and intent, complemented by an ensemble approach. The authors applied statistical significance tests to investigate the correlation between permissions and intent and found statistical evidence of a strong correlation between permissions and intent that could be leveraged to detect malware applications.

Rana et al. [20] proposed and evaluated various machine learning algorithms by applying an ensemble-based learning approach to identify Android malware associated with a substring-based classifier feature selection (SBFS) strategy. They used the DREBIN dataset and achieved better results with an ensemble learning approach.

Given that the ensemble method combines the results of multiple machine learning algorithms, it can improve the prediction performance considerably. We construct a stacking ensemble learning framework composed of five base learners and a meta classifier to achieve effective and efficient malware detection. The stacking ensemble method utilized in our paper uses all of the training data to perform k-fold cross-validation for multiple models in the first layer, so that each model has a predicted value, and then it trains the models in the second layer with these predicted values as new features.

## 3. Dataset

This study explored the statuses of Android malware applications, so we constructed a real-world Android malware dataset. From 2019 to 2020, we collected a total of 1664 Android apps in 15 categories from 4 app markets in China: Huawei application market (https://appstore.huawei.com/ accessed on 30 November 2019), Xiaomi application market (https://app.mi.com/ accessed on 1 February 2020), 360 application market (https://ext.se.360.cn/ accessed on 2 March 2020), and the Wandoujia application market (https://www.wandoujia.com/ accessed on 5 May 2020), including 806 malicious apps and 858 benign apps (Table 1). The 15 categories were finance and economics, home and life, chatting and socializing, travel and transportation, photography and videography, fashion and shopping, practical tools, video and music, sports, book reading, efficient office, news and information, learning and education, health care and entertainment. We identified whether the collected Android applications were malicious or benign by using the VirusTotal web tool, which was an online malware detection tool that used multiple anti-virus engines to detect uploaded APK files, and returned detection reports to determine whether the files were infected by viruses, worms, trojans, and various types of malware [21]. We marked APK files according to the number of engines in the report provided by VirusTotal, when an APK file was detected as malicious by at least three engines, it was marked as malicious, otherwise it was benign. The specific features of the dataset are shown in Table 2.

## 4. MFDroid

In this section, we introduce the proposed Android malware detection framework in detail, as is shown in Figure 1, which is divided into three stages: feature preprocessing, feature selection, and stacking ensemble method.

### 4.1. Data Preprocessing

We preprocess the data by utilizing AndroPyTool, a tool for extracting static and dynamic features from APK, which combines various well-known Android application analysis tools such as DroidBox, FlowDroid, Strace, and AndroGuard. AndroPyTool uses the above tools to perform pre-static analysis, static analysis, and dynamic analysis on APK files, and generate feature files in JSON format [9]. As shown in Figure 2, the first step is to check whether the sample is a valid APK file and filter out the corrupted APK files that cannot be decompiled. The second step uses AndroPyTool to extract the feature information of each sample and store it in JSON format for subsequent feature extraction and vectorization. The third step uses regular expressions to extract features, such as permissions, opcodes, and API calls from the JSON file generated by AndroPyTool. The last step is to vectorize the extracted features. If a certain bit of the feature vector is 0, it means that the feature does not appear in the APK file, and if it is 1, the opposite is true. A total of 3547 features, including permissions, opcodes and API calls, were mapped to the vector space for subsequent feature selection. Detailed feature information is shown in Table 3.

### 4.2. Feature Selection Methods

In this section, we elaborate on the seven feature selection algorithms utilized by MFDroid and introduce the feature selection strategy used in this study.

#### 4.2.1. Feature Selection Algorithms

The tree-based feature selection method is an information gain algorithm. The principle is that—the more the same types of leaf nodes are contained in a feature, the more significant the feature is in training. The tree model can calculate the feature value through learning and training importance and calculate the feature contribution index, so it can be used to remove irrelevant features. In this study, we used three tree-based feature selection algorithms: decision tree (DT), gradient boosting decision tree (GBDT), and extra-trees (ET).

Chi-square test (Chi2) is a hypothesis testing method in which the distribution of a statistic approximately obeys the chi-square distribution when the null hypothesis is established, and it is used to determine whether two variables are independent [22]. The correlation between categorical variables can be judged by calculating the chi-square value of two categorical variables. The larger the chi-square value, the greater the relationship between the two categorical variables and the lower the independence; otherwise, the smaller the relationship between the two categorical variables, the higher the independence. When the chi-square value reaches 0, it means that the factors are the same.

Genetic algorithm (GA) is a search and optimization technology inspired by the biological evolution process [23,24]. Based on the rule of survival of the fittest, it searches for the optimal solution through various genetic operations. The population in the genetic algorithm represents the possible solution set of the problem, and a population is composed of a certain number of individuals encoded by genes, and each individual corresponds to a possible solution to the problem. The genetic algorithm will randomly generate the initial population according to different coding methods, and iteratively evolve according to the principle of survival of the fittest until a suitable solution set is found.

The support vector machine based on recursive feature elimination (SVM-RFE) feature selection algorithm is a backward recursive elimination feature selection algorithm that uses the classification performance of the support vector machine as the feature selection evaluation standard, and has high recognition performance [25]. The SVM-RFE method removes the features with the smallest ranking coefficient through continuous iteration, then uses SVM to retrain the remaining features to obtain new feature rankings, and finally obtains a sorted list of features. Using the sorted list, define several nested feature subsets $F_{1}\subset F_{2}\subset \ldots \subset F$ to train the SVM, and evaluate the pros and cons of these subsets with the SVM prediction accuracy rate, to obtain the optimal feature subset.

The degree of correlation between different variables can be measured by mutual information (MI) [26]. The smaller the mutual information, the less common information, and the more independent the two variables are; on the contrary, the more common information, the more related the two variables are. According to the definition of mutual information, if and only when two random variables are independent of each other, the mutual information is 0; when the two variables are highly correlated, the mutual information of the two variables will also be large.

#### 4.2.2. Union of Feature Selection Results

On the feature vector space after data preprocessing, seven feature selection algorithms, including ET, GBDT, DT, Chi2, GA, SVM-RFE, and MI were applied. We extracted the feature index dictionary stored by each feature selection algorithm, took the union to combine the selection results of these single feature selection algorithms, stored the combined result as a dictionary, and selected from the original feature vector space according to the feature index in the dictionary. The final feature set was used in this study.

As shown in Table 4, each feature selection algorithm selected a different number of features. Therefore, the single feature selection algorithm has limitations, such as some important features may be omitted. This study utilizes the union strategy to merge the results of the 7 feature selection algorithms, and finally 2376 features remain, which are feed to the model for training. In addition, we also use the intersection strategy to test, and the results show that the feature set generated by the intersection strategy was not effective in the experiment. Because different feature selection algorithms have different internal mechanisms, the results of each feature selection algorithm are quite different. Therefore, it is necessary to choose union strategy.

### 4.3. Stacking Ensemble Method

#### 4.3.1. Base Learners

This study used SVM, GBDT, XGBoost, LightGBM, and CatBoost as base learners, and used the new feature set generated by the union strategy to train each base learner to guarantee accuracy.

SVM is often used to solve binary classification problems [27]. Given a sample training set $\left(x_{i},d_{i}\right)$, *i* = 1, 2, …, *N*, $x_{i}\in R^{n}$, $d_{i}\in \left\{\pm 1\right\}$, the principle of SVM is to try to find a hyperplane (w·x)+b=0,$\;x,w\in R^{n}$, b∈R, so that the hyperplane satisfies the classification requirements; that is, find the optimal value of the weight vector *w* and the bias *b*, so as to satisfy the formula:(1)$$d_{i}\left(w^{T}x_{i}+b\right)\geq 1-\xi_{i}\;\;\;\;i=1,2,\ldots ,N$$
where slack variable $\xi_{i}\geq 0,i=1,2,\ldots ,N$.

GBDT is a gradient boosting integration method, which is a method that adopts boosting technology. The main idea of GBDT is to follow the gradient descent direction of the loss function of the previously established model every time a model is built. If the classifier model can be built in the direction that the loss function continues to decrease, the performance of the classifier model improves. By minimizing the loss function L(θ), parameter *θ* is obtained, and its calculation formula is shown in the formula.
(2)$$\theta =\theta -\alpha \cdot \frac{\partial }{\partial \theta }L\left(\theta \right)$$

In the *m*-th iteration of GBDT, the classifier parameters of the first *m*-1 iterations are fixed. Therefore, in the *m*-th iteration, only the loss function of the *m*-th classifier needs to be minimized to obtain the corresponding classifier.

The XGBoost algorithm is an evolutionary form of the GBDT algorithm [28]. Its base learner usually selects a decision tree model and generates a new tree through continuous iteration to learn the residual between the actual value and the predicted value of all current trees and accumulate the results of all trees as the result, to get the highest possible classification accuracy. The objective function of the XGBoost algorithm is shown in Formula (3):(3)$$obj=\overset{n}{\underset{i=1}{\sum }}l\left(y_{i},{\hat{y}}_{i}\right)+\overset{K}{\underset{k=1}{\sum }}\theta \left(f_{k}\right)=\overset{n}{\underset{i=1}{\sum }}\left[f_{t}\left(x_{i}\right)g_{i}+\frac{1}{2}{\left(f_{t}\left(x_{i}\right)\right)}^{2}h_{i}\right]+\theta \left(f_{t}\right)$$
where *g_i_* and *h_i_* are the first and second derivatives of the loss function, respectively, and $\theta \left(f_{t}\right)$ is the structure of the *t*-th tree.

The LightGBM algorithm is another evolutionary form of the GBDT algorithm [29]. The algorithm uses a depth-limited leaf-wise strategy to find the node with the largest gain value from the current leaf nodes to split; it limits the depth of the tree to prevent overfitting and reduces the time to find the optimal depth tree. At the same time, when the number of splits is the same, the error can be reduced and a higher precision can be obtained. In the process of building a tree, the process of finding the optimal splitting node is the most wasteful in terms of time and computer resources. For this, LightGBM uses histogram algorithm, gradient based one-side sampling, mutually exclusive features, and the binding algorithm (exclusive feature building, EFB) to improve operating efficiency.

CatBoost is a machine learning framework based on GBDT, which is designed to solve the discrete feature problem existing in GBDT [30]. CatBoost uses the oblivious tree as the base predictor, and the index of the leaf node of the oblivious tree is converted into a binary vector whose length is the depth of the tree. CatBoost performs binarization operations on various types of features and statistical information, and then inputs the binary features into the model to calculate the predicted value. The binary features are stored in vector *B*, and the value of leaf nodes is stored in a vector of size 2*^d^*, *d* is the depth of the tree. For sample *x*, its binary vector representation is:(4)$$\overset{d-1}{\underset{i=0}{\sum }}2^{i}\cdot B\left(x,f\left(t,j\right)\right)$$
where B(x,f) represents the binary-type feature value of the sample *x* obtained from the vector *B*, and f(t,j) represents the numerical value of the binary feature in the *t*-th tree.

#### 4.3.2. Meta-Classifier

In our study, logistic regression is used as a meta-classifier, and logistic regression is proposed to deal with classification problems. Traditional linear regression only implements regression, and its basic form is as follows:(5)$$y\;=\omega^{T}x+b$$

The predicted value produced by linear regression is a series of real values, but to solve a classification problem, the predicted value should be 0 or 1, so it is further transformed with the help of the continuous function Sigmoid:(6)$$y\;=\frac{1}{1+e^{-x}}$$

The sigmoid function can convert the prediction result of linear regression into the conditional probability that the sample belongs to a certain category, and then with the help of the preset classification threshold, the classification result can be obtained. For the two-class problem, by incorporating *b* into the weight vector *ω*, the probability that the sample belongs to the positive and negative classes can be obtained:(7)$$P\left(Y=1|x\right)=\frac{exp\left(\omega \cdot x\right)}{1+exp\left(\omega \cdot x\right)}$$
(8)$$P\left(Y=0|x\right)=\frac{1}{1+exp\left(\omega \cdot x\right)}$$

Equations (7) and (8) are logistic regression models. According to Equation (7), when the predicted value of the linear function tends to positive infinity, the probability value of the predicted sample being a positive class is close to 1, otherwise it is close to 0.

#### 4.3.3. Ensemble of Classifiers

The stacking ensemble learning prediction method is shown in Figure 1. The base learner is composed of each single prediction model. First, the original dataset is used to train multiple base learners. In the training process, to reduce the risk of model overfitting, *K*-fold cross-validation is generally used to train base learners. Then, the prediction results of the base learner are formed into a new dataset, and then the secondary learners are trained to obtain the final prediction results [18]. The specific steps of the algorithm are as follows:Divide the original dataset into two parts: the original training set *D* and the original testing set *T*.Perform *K*-fold cross-validation on the base learner: randomly divide the original training set *D* into *K* equivalents $\left(D_{1},D_{2},\ldots ,D_{k}\right)$, and each base learner uses one of them as the *K*-fold test set, and the rest the *K*-1 copies are used as the *K*-fold training set. Each base learner is trained using the *K*-fold training set, and predictions are made on the *K*-fold test set, and the prediction results of each base learner are combined as the training set $\tilde{D}$ of the secondary learner.Each base learner makes predictions on the original test set *T*, and the prediction results are averaged as the validation set $\tilde{T}$ of the secondary learners.The secondary learner obtains the new dataset generated from the base learner: the training set $\tilde{D}$ and the validation set $\tilde{T}$, and then performs learning and training, and outputs the final prediction result.

The stacking ensemble learning prediction method uses *K*-fold cross-validation to reduce the risk of overfitting of the model and uses the prediction results of multiple base learners to perform secondary training (see Algorithm 1). This method can overcome the limitations of a single learner and integrate variously; it is a machine learning method that improves the accuracy and generalization of prediction results. 

In stacking the ensemble learning prediction method, the base learner can choose a strong learner, and the secondary learner can choose a simple learner, which could make the fusion effect better and avoid overfitting.
**Algorithm 1** Stacking with *K*-fold Cross ValidationInput: training data D={Xi,yi}i=1m (Xi∈ℝn,yi∈Υ)Output: an ensemble classifier *H*1: Step 1: adopt cross validation approach in preparing a training set for second-level classifier2: Randomly split D into *K* equal-size subsets: D={D1 ,D2,···,DK}3: for *k* = 1 to *K* do 4:    Step 1.1: learn first-level classifiers5:    for *t* = 1 to *T* do6:        Learn a classifier hkt from D\DK7:    end for8:    Step 1.2: construct a training set for second-level classifier9:    for Xi∈DK do10:       Get a record {Xi′,yi}, where Xi′={hk1(Xi),hk2(Xi),···,hkT(Xi)}11:   end for12: end for13: Step 2: learn a second-level classifier14: Learn a new classifier h′ from the collection of {Xi′,yi}15: Step 3: relearn first-level classifiers16: for *t* = 1 to *T* do17:   Learn a classifier ht based on D18: end for19: return H(X)=h′(h1(X),h2(X),···,hT(X))


### 4.4. Evaluation Metrics

To train the model, we split the data, of which 75% was used for training and 25% for testing. To evaluate the performance of MFDroid, some evaluation criteria were introduced, such as confusion matrix, accuracy, precision, recall, F1-score, and ROC curves. In this study, the binary classification machine learning results were classified by the confusion matrix, as shown in Table 5. The confusion matrix includes information about the predicted classification results based on machine learning and the actual classification results.
(9)$$Accuracy=\frac{TP+TN}{TP+FP+TN+FN}\;$$
(10)$$\begin{array}{c} Precision=\frac{TP}{TP+FP} \end{array}$$
(11)$$\begin{array}{c} Recall=\frac{TP}{TP+FN} \end{array}$$
(12)$$F1-score=2\times \frac{Precision\times Recall}{Precision+Recall}$$
where *TP* is the amount of malware apps correctly detected, *FP* is the amount of benign apps wrongly predicted as malware, *FN* is the amount of malware cases misclassified as benign software, and *TN* is the amount of benign apps correctly detected.

## 5. Experiments and Results

### 5.1. Dataset Analysis

In this section, we analyze different types of features. Evaluating datasets from different feature perspectives is a useful mechanism to identify differences in features between malicious and benign applications and to draw general conclusions.

#### 5.1.1. Permissions Required Analysis

Table 6 shows the top 20 permissions used by malicious applications and benign applications. Most samples of different types need to apply for INTERNET, ACCESS_NETWORK_STATE and WRITE_EXTERNAL_STORAGE permissions, so there is no obvious difference. The WRITE_SETTINGS permission allows programs to read or write system settings. Moreover, 79% of malicious apps have applied for this permission, and 64% of benign apps have applied for this permission. In particular, our research analyzes 24 dangerous permissions officially announced by Android and counts the application frequency of dangerous permissions in malicious applications and benign applications. As can be seen from Figure 3, there are three permissions related to SMS messages, the SEND_SMS permission allows the application to send SMS messages, the RECEIVE_SMS permission allows the application to receive SMS messages, and the READ_SMS permission allows the application to read SMS messages. These three permissions show obvious differences in different types of applications, and the frequency of malicious applications applying for SMS message-related permissions is significantly higher than that of benign applications. As we all know, some malicious applications may cause serious property damage or privacy leakage to users by stealing SMS messages, such as verification codes and other private messages. At the same time, the PROCESS_OUTGOING_CALLS permission, which allows applications to view the number being dialed during an outgoing call and can choose to redirect the call to another number or abort the call entirely, is significantly more frequently applied in malicious apps than in benign ones. The official Android documentation marks its protection level as dangerous, and it can be used to access very sensitive information.

It is worth noting that, whether it is ordinary permissions or dangerous permissions, the frequency of permission applications for malicious applications and benign applications is not significantly different, and further analysis shows that the use of specific permissions is also very similar. Therefore, it is difficult to achieve the detection effect in the past only by relying on permissions as a feature of machine learning algorithms. More fine-grained API calls, Opcode, etc., are required as features to participate in the detection. Various types of application features are conducive to improving the performance of detection.

#### 5.1.2. API Calls Analysis

The mechanisms inherent in API calls make it possible to identify relevant differences between applications with good or illegal intents. In this study, we selected some typical API calls for statistical analysis according to the common attack types of Android malware. As shown in Table 7, among the Android/telephony/TelephonyManager API calls, 87% of malicious applications call this package, on the contrary, only 46% of benign applications call this package. In permissions analysis, we observed that phone-related services were used more frequently in malware samples.

#### 5.1.3. Opcodes Analysis

In this subsection, we examine the use of opcodes. Table 8 is the top 20 opcodes used by malicious applications and benign applications. Typically, an opcode-based analysis of the frequency of use does not yield a relevant conclusive assessment. Our dataset demonstrates that there is no large difference between the top 20 frequently used opcodes for malicious and benign applications. However, opcodes are instructions executed by applications that must be used by all applications to perform scheduled activities. Therefore, it is of great significance to take opcode as one of the characteristics of detecting Android malware.

### 5.2. Analysis of Feature Selection Results

Our study used eight feature selection methods, namely ET, GBDT, DT, Chi2, GA, SVM-RFE, MI, and non-selection (NS), experiments with MFDroid, each feature selection algorithm selected a different number of features. It can be seen from Figure 4 and Table 9 that the experimental results of the single feature selection algorithm are not significantly different (see Figure 5 and Figure 6). It is worth noting that the number of features selected by ET is small but its experimental results are high. GA has the largest number of features selected among the seven feature selection algorithms, and the experimental results are the best. Considering the limitations of the single feature selection algorithm, some important features may be missed. Therefore, this study adopts the union strategy to merge the results of the seven feature selection algorithms, and finally retains 3547 features and sends them to the model for training. Experiments show that the F1-score of the union strategy reaches 96.0%, which is 2.3% higher than that of GA, and the accuracy value is 2.4% higher. Therefore, the feature selection method using the union strategy is accurate and effective.

To compare our work more easily with other existing studies, we introduce the OmniDroid dataset to test our model (see Figure 7, Figure 8 and Figure 9). As shown in Table 9, our adopted feature selection union strategy outperforms other feature selection algorithms and our model outperforms other comparative methods on OmniDroid. Compared with [31], our method performs better on the static features of OmniDroid.

### 5.3. Detection Performance Evaluation of MFDroid

To verify the effect of the multi-model fusion framework MFDroid proposed in this study in the field of Android malware detection, combined with the five-fold cross-validation method, SVM, GBDT, XGBoost, LightGBM, CatBoost, and MFDroid proposed in this study were compared. The experimental results are shown in Figure 10 and Table 10.

Through comparison, it is found that the evaluation index of MFDroid is significantly higher than that of other single models. Compared with SVM, GBDT, XGBoost, LightGBM and CatBoost, the F1-score of MFDroid is increased by 4.9%, 7.4%, 4.6%, 3.6%, and 3.0%, and the accuracy value is increased by 5.1%, 7.9%, 4.8%, 3.8% and 3.1%, respectively. It was proven that MFDroid has high accuracy and stability in the field of Android malware detection (see Figure 11 and Figure 12).

## 6. Conclusions and Future Work

We built a real-world Android malware dataset and proposed an Android malware detection framework based on stacking ensemble learning, called MFDroid, to accurately and effectively detect the application software on the Android operating system, and filter out the malware, to maintain the software quality of the Android application market and the information security of users. First, the framework combined the results of seven feature selection algorithms to obtain a new feature set. Then, combined with the advantages and characteristics of various learners, we set five base learners in the first layer of the stacking ensemble learning framework, and the second layer used logistic regression as a meta-classifier. The results showed that MFDroid was an effective Android malware detection framework.

Regarding the limitations of a single feature selection algorithm, we conducted experiments on each feature selection algorithm, and the performance index of our method was higher than that of a single feature selection algorithm, which also indirectly proves that a single feature selection algorithm will miss some features. Regarding the limitations of a single machine learning algorithm—a single machine learning algorithm will be affected by data skew, and the computational overhead is relatively large. In addition, it may be affected by the initial setting, is sensitive to noise, and cannot handle high-dimensional features well. In the future, we will explore static and dynamic features that can describe application behavior more comprehensively, by using multiple types of features to improve detection accuracy.

## Figures and Tables

**Figure 1 sensors-22-02597-f001:**
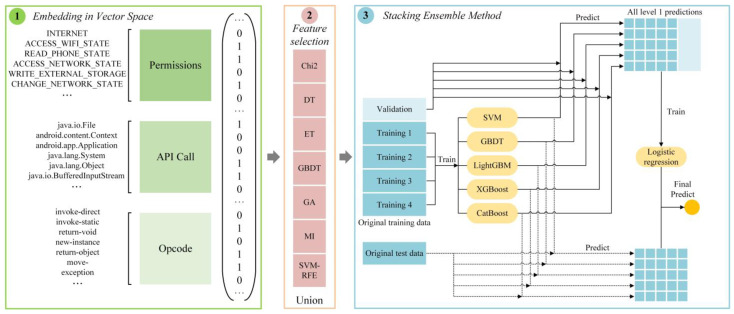
The architecture of MFDroid.

**Figure 2 sensors-22-02597-f002:**
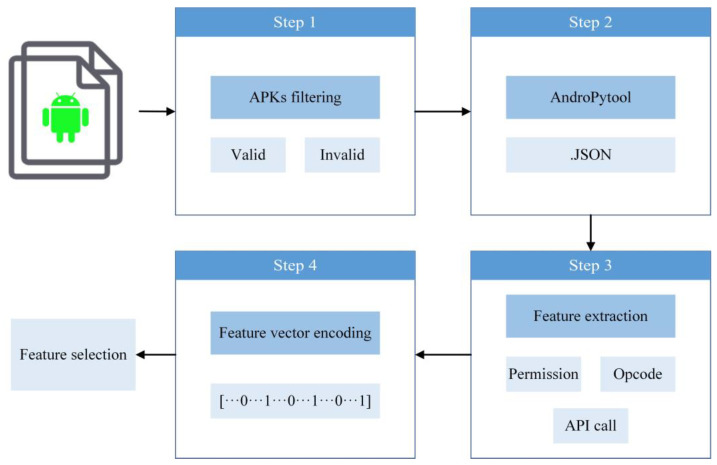
The process of data preprocessing.

**Figure 3 sensors-22-02597-f003:**
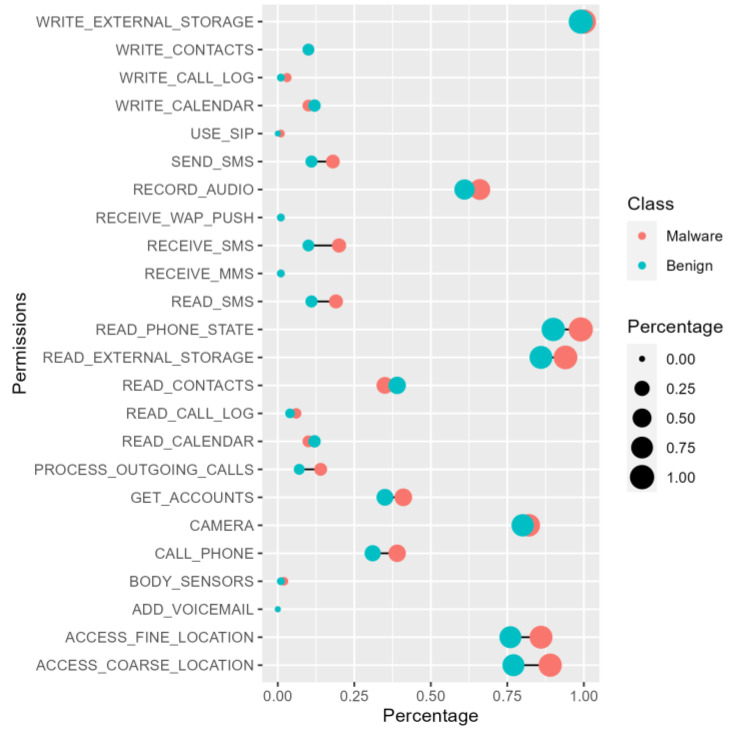
A total of 24 dangerous permissions in the percentages of malicious applications and benign applications, respectively.

**Figure 4 sensors-22-02597-f004:**
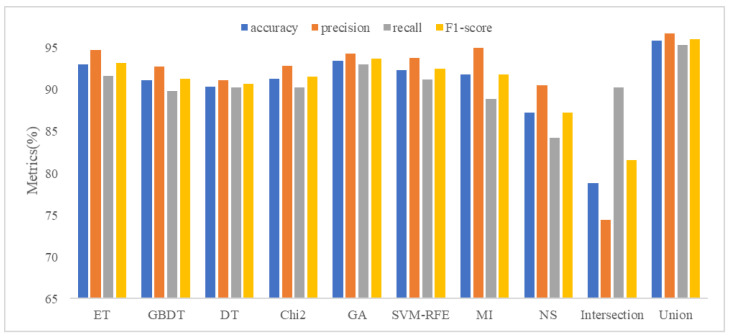
Feature selection results comparison on our dataset.

**Figure 5 sensors-22-02597-f005:**
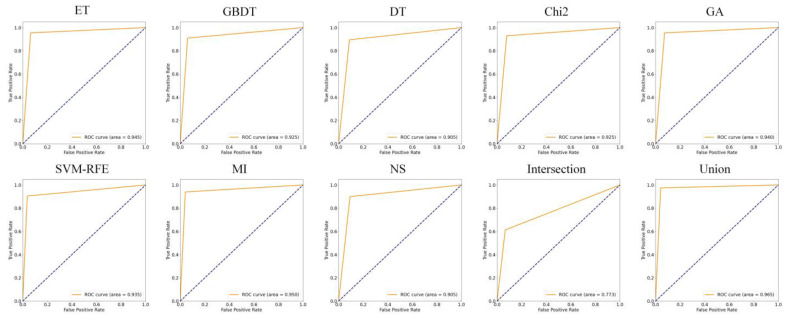
ROC curves of different feature selection methods on our dataset.

**Figure 6 sensors-22-02597-f006:**
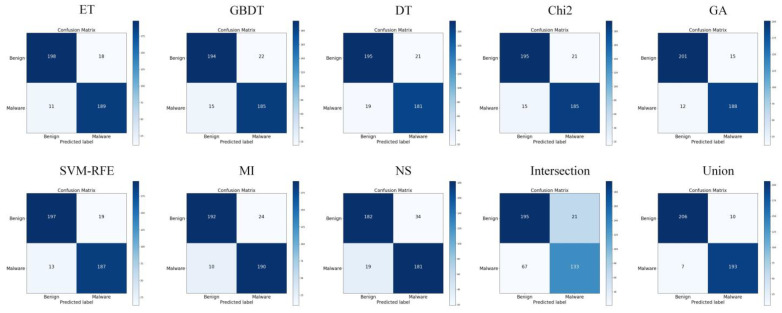
Confusion matrix of different feature selection methods on our dataset.

**Figure 7 sensors-22-02597-f007:**
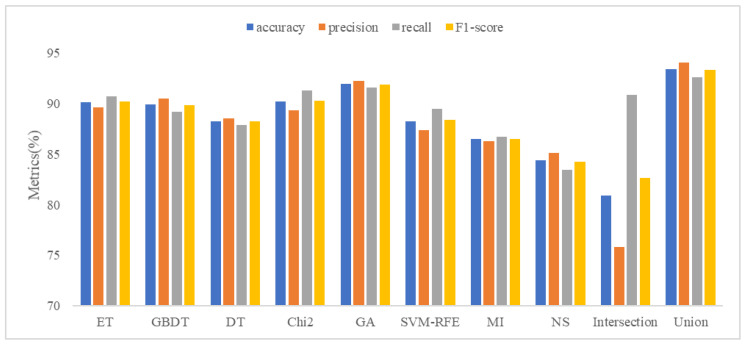
Feature selection results comparison on OmniDroid dataset.

**Figure 8 sensors-22-02597-f008:**
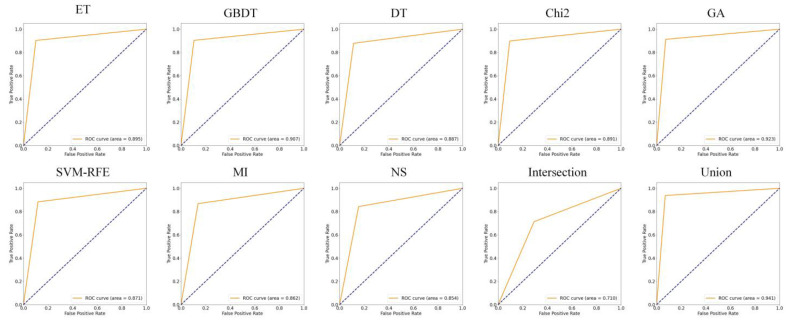
ROC curves of different feature selection methods on OmniDroid dataset.

**Figure 9 sensors-22-02597-f009:**
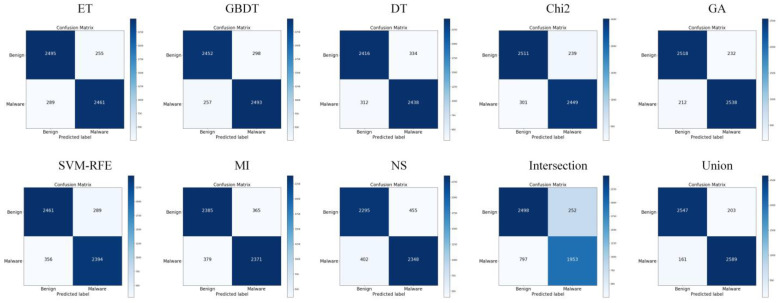
Confusion matrix of different feature selection methods on OmniDroid dataset.

**Figure 10 sensors-22-02597-f010:**
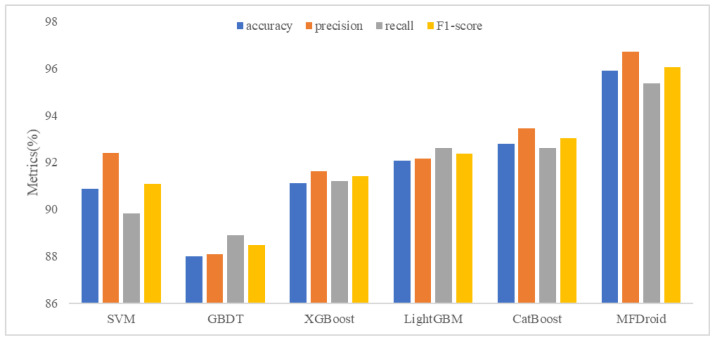
Comparison of results between MFDroid and single classifier.

**Figure 11 sensors-22-02597-f011:**
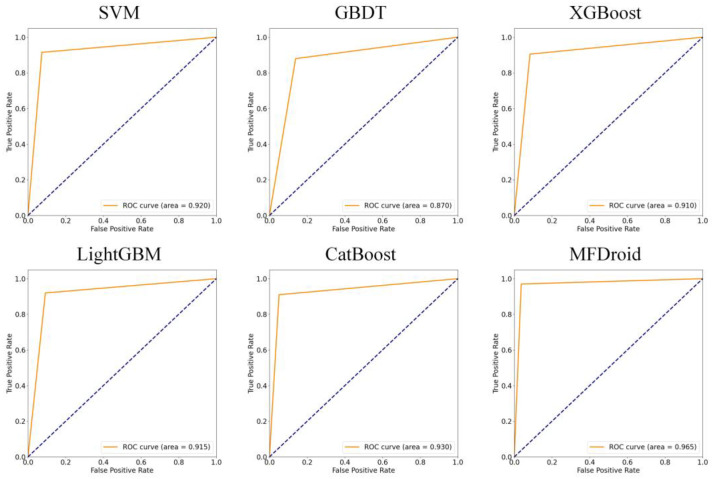
ROC curve of MFDroid and single classifier.

**Figure 12 sensors-22-02597-f012:**
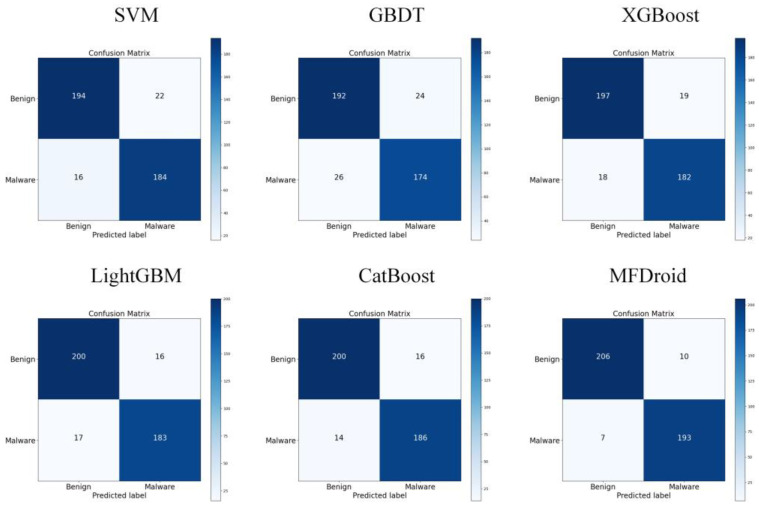
Confusion matrix of MFDroid and single classifier.

**Table 1 sensors-22-02597-t001:** Number of applications.

Category	Count
Malware	806
Benign	858
Total	1664

**Table 2 sensors-22-02597-t002:** Number of dataset features.

Feature	Count
Permission	146
Opcode	218
API call	3183
Activity	4071
Service	3219
Receiver	4633
apiPackage	212

**Table 3 sensors-22-02597-t003:** Representation of feature vectors.

Feature	Category	Amount	Combined Vector
Permission	0–1	146	The three features formed a 3547-dimensional vector
Opcode	0–1	218	
API call	0–1	3183	

**Table 4 sensors-22-02597-t004:** The number of features selected by each feature selection algorithm.

Algorithm	ET	GBDT	DT	Chi2	GA	SVM-RFE	MI
Amount	1452	1319	1500	1500	1687	1893	1366

**Table 5 sensors-22-02597-t005:** Confusion matrix.

Predicted	Actual	Positive	Negative
Positive		True Positive (*TP*)	False Positive (*F**P*)
Negative		False Negative (*FN*)	True Negative (*TN*)

**Table 6 sensors-22-02597-t006:** Top 20 most common permissions in malicious and benign apps.

Malware		Benign	
Permission	%	Permission	%
INTERNET	100%	INTERNET	99%
ACCESS_NETWORK_STATE	100%	WRITE_EXTERNAL_STORAGE	99%
WRITE_EXTERNAL_STORAGE	100%	ACCESS_NETWORK_STATE	99%
ACCESS_WIFI_STATE	100%	ACCESS_WIFI_STATE	94%
READ_PHONE_STATE	99%	READ_PHONE_STATE	90%
READ_EXTERNAL_STORAGE	94%	READ_EXTERNAL_STORAGE	86%
WAKE_LOCK	90%	WAKE_LOCK	82%
ACCESS_COARSE_LOCATION	89%	VIBRATE	81%
GET_TASKS	88%	CAMERA	80%
VIBRATE	88%	ACCESS_COARSE_LOCATION	77%
CHANGE_WIFI_STATE	87%	ACCESS_FINE_LOCATION	76%
ACCESS_FINE_LOCATION	86%	CHANGE_WIFI_STATE	72%
CAMERA	82%	GET_TASKS	71%
WRITE_SETTINGS	79%	RECEIVE_BOOT_COMPLETED	65%
SYSTEM_ALERT_WINDOW	76%	REQUEST_INSTALL_PACKAGES	64%
MOUNT_UNMOUNT_FILESYSTEMS	74%	WRITE_SETTINGS	64%
RECEIVE_BOOT_COMPLETED	72%	RECORD_AUDIO	61%
CHANGE_NETWORK_STATE	70%	SYSTEM_ALERT_WINDOW	60%
REQUEST_INSTALL_PACKAGES	67%	MOUNT_UNMOUNT_FILESYSTEMS	59%
RECORD_AUDIO	66%	CHANGE_NETWORK_STATE	58%

**Table 7 sensors-22-02597-t007:** Percentage of common API calls by attack type in malicious and benign applications.

Attacks	API Calls	Malware	Benign
Telephone	Android/telephony/TelephonyManager	87%	46%
Location	Android/telephony/gsm/GsmCellLocation	62%	34%
	Android/location/Address	80%	38%
	Android/location/Location	80%	39%
	Android/location/LocationManager	77%	39%
Camera	Android/hardware/Camera	57%	29%
Storage	Android/os/Environment	90%	45%
	Android/content/Context/getExternalFilesDir	75%	36%
	Java/Io/Bytearrayoutputstream	94%	47%

**Table 8 sensors-22-02597-t008:** Top 20 most common opcodes in malicious and benign apps.

Malware		Benign	
Opcode	%	Opcode	%
const/16	100%	const-wide/16	100%
const/4	100%	const/16	100%
goto	100%	const/4	100%
if-eqz	100%	goto	100%
if-nez	100%	if-eqz	100%
invoke-direct	100%	if-nez	100%
invoke-static	100%	invoke-direct	100%
invoke-super	100%	invoke-static	100%
invoke-virtual	100%	invoke-super	100%
move-exception	100%	invoke-virtual	100%
move-result	100%	move-exception	100%
move-result-object	100%	move-result	100%
new-array	100%	move-result-object	100%
new-instance	100%	new-array	100%
return-object	100%	new-instance	100%
return-void	100%	return-void	100%
sget-object	100%	sget-object	100%
sput-object	100%	sput-object	100%
aput-object	100%	throw	100%
const-wide/16	100%	move-object	100%

**Table 9 sensors-22-02597-t009:** Evaluation metrics for different feature selection methods on ours and OmniDroid.

No.	Algorithm	Accuracy		Precision		Recall		F1-Score	
		Ours	OmniDroid	Ours	OmniDroid	Ours	OmniDroid	Ours	OmniDroid
1	ET	0.930	0.901	0.947	0.896	0.917	0.907	0.932	0.902
2	GBDT	0.911	0.899	0.928	0.905	0.898	0.892	0.913	0.898
3	DT	0.904	0.883	0.911	0.886	0.903	0.879	0.908	0.882
4	Chi2	0.914	0.902	0.929	0.893	0.903	0.913	0.916	0.903
5	GA	0.935	0.920	0.944	0.922	0.931	0.916	0.937	0.919
6	SVM-RFE	0.923	0.883	0.938	0.874	0.927	0.895	0.925	0.884
7	MI	0.918	0.865	0.951	0.863	0.889	0.867	0.919	0.865
8	NS	0.873	0.844	0.905	0.851	0.843	0.835	0.873	0.843
9	Intersection	0.788	0.809	0.744	0.758	0.903	0.908	0.815	0.882
**10**	**Union**	**0.959**	0.934	**0.967**	0.941	**0.954**	0.926	**0.960**	0.933

**Table 10 sensors-22-02597-t010:** Evaluation metrics of MFDroid and single classifier.

No.	Model	Accuracy	Precision	Recall	F1-Score
1	SVM	0.908	0.924	0.898	0.911
2	GBDT	0.880	0.881	0.889	0.886
3	XGBoost	0.911	0.916	0.912	0.914
4	LightGBM	0.921	0.922	0.926	0.924
5	CatBoost	0.928	0.935	0.926	0.930
**6**	**MFDroid**	**0.959**	**0.967**	**0.954**	**0.960**

## Data Availability

Not applicable.

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
