# Peer review of "MFDroid: A Stacking Ensemble Learning Framework for Android Malware Detection"

_sensors, 2022, doi:10.3390/s22072597_

Round 1
Reviewer 1 Report
Overall this is a very well written paper and effectively presents the scientific methodologies.
Even with the recent advancement in design of Android OS, it still remains one of the main target for adversaries and malware developers. The authors propose MFDroid; which makes use of stacking ensemble learning for Android malware detections.
I recommend the following for the authors:
1) Please compare your work with previous related works; for example Rana, M. S., & Sung, A. H. (2020). Evaluation of Advanced Ensemble Learning Techniques for Android Malware Detection. Vietnam Journal of Computer Science, 7(02), 145-159.
2) More details about the benign dataset can add more clarity into the data used. It is crucial to emphasize on the training dataset being balanced.
3) A discussion section would be very beneficial. Although achieving a 100% precision-recall may not be possible, but authors should discuss potential future work and steps to enhance this method (for example explaining the FP/FN; could more features result in less FP/FN?)
4) Please provide a summary on potential ways to evade MFDroid.
Reviewer 2 Report
the litrture survey is very shallow and lacks many similar work
no comparison is made to the other recent work to judge the soundness of the algorithm
the dataset is personally collected, why did not use the avaliable dataset in addition to their dataset
the details of the dataet are not given, so looks like a black box to us
the size of training/testing of the dataset is not given
the ensemble algorithm should be better shown in a flowchart or a pseduo code instead of narrative
figures 4 and sven should be bar graph not continous line
Reviewer 3 Report
See attached file.

Reviewer 4 Report
Dear Authors,
Many thanks for your manuscript submission to MDPI Journal of Sensors. After my review from all related aspects, I justify it is a very good research article, it had well presented their study, displayed the quantitative results, and concluded that great efforts must be made on displayin static and dynamic features on improving detection accuracy (for both malware and benign Android software apps). My overall recommendation is an "Accept after minor revision", after fixing the required major and minor edits.
With respect to the professional standards on publishing MDPI journal, the following aspects are suggested to be considered for improvements or calibrations, which I listed (may not limited to all of those) as follows:
Major problematic issues expected to be addressed in your edited version:
1) Abstract: the length is moderate, while the last sentence in Line 19 is too generic. Some keynote concluding remarks (including your mainstream key results), are also expected to be added in your updated version. Thanks a lot!
2) While the topic of Malware detection is both original or relevant in that research field, there are two issues needs clarification. If the limitations of static analysis for malware detection was presented in Year 2007, what are the tendency and development in the following 10 years? Meanwhile, what are some of the critical progressive study in your related work, and parallel comparison to other investigations? To be frank, I think the authors need to be more clear on how your study enriched the subject area in contrast to other published articles, not just the latest approaches in the past 3 years.
3) The authors presented a large amount of well-arranged data and digits. What specific improvements (or technical innovations) you had taken into account, with respect to the proposed framework and schemes by MFDroid?
4) Section 4.2 (Feature selection methods) is a bit too generic. Meanwhile, as I specified that Sections 4-5 are filled up with facts and data analysis, since it already presented a conclusive summary on major contributions of your work (while not very specific), adding a section on discussions and findings on the research study is suggested for supplement, which should be concise and specific, including your justification on limitations of study. Thanks.
5) Conclusions: To be frank, the current version of conclusions is consistent with the evidence and arguments presented in the content, while it needs further work to address the posted mainstream topics on Android malware detection. The authors may need to supplement a paragraph on opening questions with respect to their limitations of study, potential weakness of research tools, and another paragraph to summarize the future work and prospective research orientations, which should be cohensive and coherent for the framework on Android related malware detection (not merely aiming at the data observed from latest few years, while connecting merits and main contributions in a wider range of research domain in the recent decade).
6) Figures and Tables: Stop crossing over two adjacent pages for a table. Please apply the required edits to adjust the arrangment of Tables 4, 6, 7. Meanwhile, the legends and notations of Figures 4-5 should be enlarged (better the same font size as Figures 6-7).
7) References: While the References are well-arranged, it contains the following issues: a) Forgot to comply with template formats (Abbreviated terms for citing journals), the number of volume need to be italic ; b) Low-frequency to cite keynote research study among the prior years (2008-2016), which is rarely visible; c) Some other advice: consider citing more journals on information security and highly visible ACM transactions oriented at the topic of malware detection, and check the lastest classification models based on supervised learning and semi-supervised learning, include some highly visible publications close to the study of Android malware detection in Year 2020-2022 (if already published, conduct some parallel comparison when citing these related publications).
Some minor problematic issues to be addressed in your revised version:
a) Lines 297-298: "m-1 iterations" and "m-th classifier". Please update the following "m" both in italic shape. In Subsection 4.3.3, check if the notations D, T, K, for training set D, testing set T, and K-fold, etc. need to be italic.
b) Change the title of Section 5 by "Experiments and Results" and supplement the required updates. Meanwhile, I think Section 3 could be merged into the first subsection of this section, i.e., "Datasets", "Experimental Setup", "Dataset analysis", etc.
c) Tables 7-8: consider rounding up the numerical results to 3 digits after each decimal, which is uniform to all the quantitative scores.
d) Add the discussion section after the experiments and results, and insert concluding remarks in the related part behind the corresponding locations.
e) Literal improvement should be done in your edits. A few grammatical errors still exist in this version, please fix each of them along with improving the use of English in the content. Thanks very much!
Once again, thank you for your interests on publishing article in MDPI Journal of Sensors. We look forward to seeing your future great success. Stay safe and good luck!
With warm regards,
Yours faithfully,
Round 2
Reviewer 2 Report
can be accepted
Author Response
Thank you for your valuable review comments, we have made revisions based on your comments and corrected some grammatical errors.
Reviewer 3 Report
The paper notably improved after its revision. I have just a minor suggestion. In order to augment the comparison readability between this paper results and relative results coming from similar works within the literature, I suggest the Authors to join Table 9 and Table 10. In so doing, the new Table will help the readers in appreciating the comparison.
Author Response
Thank you for your valuable review comments, which we have revised based on your comments. We merged Table 9 with Table 10, renamed it as Table 9, and deleted the original table. At the same time, we have revised the reference to the table in the article.